

# A very high-resolution assessment and modelling of urban air quality

Tobias Wolf*, Lasse H. Pettersson and Igor Esau

Nansen Environmental and Remote Sensing Center, Thormøhlens gate 47, 5006, Bergen, Norway

*Correspondence to*: Tobias Wolf (tobias.wolf@nersc.no)

**Abstract.** Urban air quality is one of the most prominent environmental concerns for modern city residents and authorities. Accurate monitoring of air quality is difficult due to intrinsic urban landscape heterogeneity and superposition of multiple polluting sources. Existing approaches often do not provide the necessary spatial details and peak concentrations of pollutants, especially at larger distances from monitoring stations. A more advanced approach is needed. This study presents a very high-resolution air quality assessment with the large-eddy simulation model PALM, capitalizing on local measurements. This fully

three-dimensional primitive-equation hydro-dynamical model resolves both structural details of the complex urban surface and turbulent eddies larger than 10 m in size. We ran a set of nine meteorological weather scenarios in order to assess the dispersion of pollutants in Bergen, a middle-sized Norwegian city embedded in a coastal valley. This set of scenarios represents typically observed conditions with high air pollution from nitrogen dioxide ($NO_2$) and particulate matter ($PM_{2.5}$). The modelling methodology helped to identify pathways and patterns of air pollution caused by the three main local air pollution

sources in the city. These are road vehicle traffic, domestic house heating with wood-burning fireplaces and ships docked in the harbour area next to the city centre. The study produced vulnerability maps, highlighting the most impacted districts for each scenario.

## 1 Introduction

Patterns of atmospheric pollution in the urban environment are rather variable and spatially heterogeneous. Air quality may
regularly deteriorate to harmful levels in the vicinity of near-surface emission sources, such as major traffic junctions and low-raised chimneys. Widely accepted statistical models provide reasonably accurate assessment of pollutant concentrations near the strong emission sources assuming availability of representative meteorological input (e.g. Isakov et al., 2017). At larger distances from the sources, or in more complex flow settings, however, their accuracy drastically deteriorates. It has been understood that pockets of polluted air could be trapped in a weakly turbulent (weakly diffusive) flow and transported over

longer distances. This turbulent transport by larger eddies cannot be successfully described by phenomenological statistical methods. Its adequate representation requires hydro-dynamic modelling where at least the largest carrying eddies would be explicitly resolved. Moreover, intricate configurations of buildings, streets and other urban surface objects create preferable pathways for wind and turbulent diffusion. Along those pathways, the pollutants disperse over much larger distances, thus, raising air quality concerns in distant districts that otherwise would be considered as unaffected.



The highest concentrations of atmospheric pollutants are often found under persistent calm and cold weather conditions. This is also the case for the coastal city of Bergen, Norway, that is subject to this study. Such conditions are characterized by a stably stratified lower atmosphere and strongly suppressed turbulent diffusion (Davy and Esau, 2016; Zilitinkevich and Esau, 2005). In the most extreme cases, known as temperature inversions, the air temperature is increasing with height (Wolf et al.,

2014) trapping the turbulence, and therefore, pollutants, in a shallow layer near the surface. Even in such conditions there could be non-negligible turbulence as well as horizontal transport of pollutants driven by local circulations (Wolf-Grosse et al., 2017a). If the calm weather conditions persist sufficiently long, concentrations of air pollutants may reach levels in excess of legal thresholds for air pollution (Bergen Kommune, 2019; European Commission, 2019), while their spatial pattern would be highly heterogeneous.

Local air quality is frequently assessed with simplified statistical models, such as a family of Gaussian models, e.g. CALINE, or with more sophisticated models, which include parametrized turbulent diffusion, such as e.g. AIRMOD (Daly and Zannetti, 2007). A set of models recommended by the US and European environmental protection agencies could be found correspondingly on EPA (2019) and on EEA (2019). Statistical models are poor in predicting horizontal pollution transport in turbulent atmospheric boundary layers (ABLs) as they do not account for turbulent eddies, meandering flows and flow-surface

structure interactions (Sun et al., 2016). In recent years, computational fluid dynamics models have been tried in assessments of the urban and road-level dispersion. As an example, one may mention simulations by Steffens et al. (2014) of the wind tunnel experiment conducted by Heist et al. (2009). This study investigated concentration gradients of a tracer gas under twelve different roadway configurations. It concluded that near-road structures impact dispersion of pollutants near roadways, and therefore, atmospheric modelling is needed to design barriers to control the impact of vehicular emissions.

Recent advances in computational fluid dynamics and growing performance of parallel computers open an opportunity to further extent the model-based urban air quality assessment. Turbulence-resolving, or at least turbulence-permitting, large-eddy simulation models have been already used in several cities to investigate turbulent flows and atmospheric pollution (e.g. Castillo et al., 2009; Gronemeier et al., 2017; Keck et al., 2014; Letzel et al., 2008; Park et al., 2015; Resler et al., 2017). Cécé et al. (2016) used a turbulence-permitting model to study air quality (nitrogen oxides, NOx) in a coastal mountain area of a

tropical island where local circulations are well developed and influential. A consistent description of the circulation and pollution effects in a coastal mountain valley can be found in Fernando et al. (2010). These and other studies convinced us that the large-eddy simulation models could accurately resolve the dispersion of pollutants even in a complex environment without the current need for potentially unsuitable statistical fitting.

This study makes the next step on the bridge between idealized feasibility studies and applied air quality assessments. It utilizes

the Large-Eddy Simulation (LES) Model PALM (Maronga et al., 2015) to investigate the dispersion of pollutants in a weakly turbulent ABL under archetypical, but frequently observed, weather conditions, which lead to dangerous deterioration of the air quality, in our case in Bergen. Bergen is embedded in a relatively deep and narrow valley ending in a large ocean fjord. The minimum distance between the mountains is approximately 1 km when measured across the valley floor; it is approximately 4 km when measured between the mountain peaks. The polluted air during cold winter days tends to accumulate





and stagnate in the valley, whereas local circulations redistribute the pollutions across the central city districts. Thus, the local circulations are likely to determine air quality for the districts' populations. The effect of the local circulations could be accounted for in the PALM simulations (Wolf-Grosse et al., 2017a), but not in statistical models relying on coarse spatial resolution (~ 1 km) mesoscale models.

In this study, the dispersion of nitrogen dioxide ($NO_2$) and small-fraction particulate matter ($PM_{2.5}$) is modelled over the densely populated central Bergen under weak winds and typical scenarios of measured severe air pollution. As such, this study presents a necessary element of a an integrated urban pollution assessment and warning system envisaged in (Baklanov et al., 2007). The chosen approach could be useful for the design of high-resolution climate information services assessing informed decision-making at the municipal level (e.g. Bauer et al., 2015; Letzel et al., 2012; Ronda et al., 2017). Results from this study
have already been adopted by the Bergen Harbour Authority (BOH) to assist their routine assessment of the impact of ship exhaust from the harbour.

The manuscript has the following structure. The next section describes the local geographical and data context for the city of Bergen. The third section presents the modelling approach with PALM and the analysed meteorological scenarios. The fourth section presents the results of this study. The fifth section provides a broader discussion with generalizations of the
methodology, data usage and policy implications. The final section briefly summarizes the conclusions.

## 2. Data

### 2.1 Site description

The western coast of Norway is known for its picturesque mountain landscapes with sea inlets (fjords) penetrating deep into coastal valleys. Similar, if not as dramatic, settings with coastal valleys opening into sea inlets and bays are frequently
accommodating harbour cities in other parts of the globe as well. Therefore, as we believe, the methodology and experience described in this study might be of interest to the research and urban management communities worldwide.

Bergen is the second largest city in Norway. The city is located at 60.4°N and 5.3°E. The mountains around the city have peak elevations between 284 m and 643 m above sea level. They protect the valley from storms, significantly reducing the surface layer wind speed (Jonassen et al., 2013). The northern location with its weak solar irradiation during wintertime and cold air
pooling in lower parts of the relief causes frequently observed but highly local temperature inversions during periods with persistently calm and clear weather. These inversions can last through several days. The resulting weak turbulent mixing of the lower valley atmosphere fosters the accumulation of locally emitted pollutants despite only moderate emissions. A climatology of temperature inversions and air pollution events in Bergen was previously reported in Wolf et al. (2014).

Bergen has more than 275,000 inhabitants. More than 75,000 of them reside in the central districts, located in the elongated
central Bergen valley, which is in focus of this study. The valley opens toward a sea inlet (Byfjorden) in the northwest. It widens towards a large brackish water lake and more residential areas in the southwest. Figure 1 shows the relief of the studied area, the simulation domain, as well as the location of the major air pollution sources in the city.



A reader may would like to know that the air quality in Bergen is monitored continuously since 2002 with an increasing number of measurement stations. In addition, a routine air quality forecast exists within the "Bedre Byluft" national project for monitoring and prediction of air pollution in Norwegian cities. The results from the measurements are available under (NILU, 2019). The predictions can be found under (Miljødirektoratet, 2019). Both, measurements and predictions are regularly

summarised (Bergen Kommune, 2018; Tarrasón et al., 2017). The forecast system has recently been changed. It was based on the AirQuis/EPISODE air quality model using meteorological input from a 1 km mesoscale numerical weather prediction (NWP) model specifically run for this application. The new forecast system is based on the "Nasjonal Bergningsverktøy" (NBV) tool. The NBV uses uEMEP as a dispersion model and the control run of the standard METCoOp Ensemble Prediction System (MEPS) for Norway with a 2.5 km spectral resolution as the meteorological input (Denby and Süld, 2015). With this,

the previous AirQuis/EPISODE model at least to some degree included the valley topography into the dispersion calculations via a higher resolution of the meteorological input and a meteorological pre-processor that created a divergence-free wind field in the valley (Baklanov et al., 2007; The World Bank, 2009).

## 2.2. Data sets

This study demonstrates the high-resolution modelling methodology using the comprehensive local data context. The model

simulations require high-resolution topographic data, emission inventories for the different sources of pollution, and vertical profiles of the large-scale (geostrophic) wind and temperature. All data sources are summarized in Table 1. Their detailed description is presented below.

The added value of the high-resolution model simulations is created by their ability to resolve the local relief features, which control the air flow and the turbulent dispersion. We run the PALM model with 10 m spatial resolution. To obtain the adequate

relief, we used a topographic laser-based data set as described in Wolf-Grosse et al. (2017a). These topographic data were processed to ensure consistency and eliminate artefacts in the airborne data. The processed data constitute a Digital Elevation Model (DEM) for the Bergen municipality. However, this DEM does not cover the entire simulation domain, as we run the simulations over a significantly larger area to resolve multiple local flows steered by the topographic relief. We extended the DEM adopting Digital Surface Model (DSM) data. The DSM does not include buildings, but outside the urbanized central

districts, and sufficiently far away from the focus area, we assume that it is sufficient to resolve the major features of the relief and coastal line.

We selected the typical air pollution scenarios for this study based on joint air quality and meteorological data analysis. Routine air pollution measurements since 2003 are available from two of the local air pollution stations, namely, from the Danmarksplass (DP) and Rådhuset (RH) stations in the central Bergen districts (see Fig. 1). The DP station represents an area

affected by heavy road traffic, whereas the RH station serves as an urban background reference. We jointly analysed the air quality and meteorological records for several pollution episodes, when high concentrations of $NO_2$ and $PM_{2.5}$ were observed. As local meteorological data we used measurements from the automatic weather stations located at the Florida neighbourhood (on top of the Geophysical Institute), the Ulriken summit, and in the Jekteviken and Skolten harbour areas. For the large-scale





weather conditions we used data from the retrospective meteorological analysis ERA-Interim (Dee et al., 2011), and from the local boundary layer temperature profiles observed by a microwave radiometer MTP-5HE, which the Nansen Center has operated on top of the Geophysical Institute since 2011.

The central Bergen districts are affected by emissions from three major polluting sources: vehicles on the road network; ships in the Bergen harbour; and wood-burning fireplaces in private residences. The road network is contributing to both, emissions of $NO_2$ and $PM_{2.5}$. To specify these emissions, we used the gridded traffic density per road segment. It was available from registers at the national road authority for the main streets and from a traffic model run by the national road authority for the side-streets.

Emissions from ships at berth in the harbour are contributing to both, emissions of $NO_2$ and $PM_{2.5}$. The Bergen harbour is located in the historic city centre, and therefore, ships docked at berth in the harbour can directly contribute to the local air pollution. Here we considered only emissions from supply-vessels for the offshore oil industry, during their periods at berth in Bergen harbour. The impact of these ships being docked right in the city centre of Bergen to local air pollution is still highly disputed by the local population. The BOH provided us with data to specify the ships' location in the harbour between January 2015 and March 2016. For the emission rates per ship we used the emission factors from the certification documentation of two representative ships.

Domestic heating with wood-burning fireplaces only contributes to emissions of $PM_{2.5}$. We assessed these emissions through accounting for estates with registered active fireplaces. These data were provided by the Bergen Fire Department. For the emission rates per fireplace we used typical emission factors for the existing mixture of new, clean burning wood ovens and older ones with higher emissions. All three emission sources are graphically summarised in Figure 1.

## 3. Model

### 3.1 The PALM Model

The high-resolution air-quality assessment requires a proper atmospheric model, which is able to resolve the most energetic turbulence eddies in fully three-dimensional simulations. This ability to resolve the turbulent motions and interactions across a multitude of scales distinguishes the turbulence-resolving, or at least turbulence-permitting, LES models from more traditional meteorological and cloud-resolving models. The traditional meteorological models have the whole turbulence spectrum collapsed in their closure schemes, even if those models are running at 1 km or sometimes higher resolutions. The turbulence closure schemes are designed for weakly stratified, horizontally homogenous ABLs over a flat terrain. Their application to a strongly stably stratified ABL commonly results in excessive vertical mixing, and hence, in erroneous pollutant transport. The results of the GABLS inter-comparison exercises are instructive to one working with this issue (Cuxart et al., 2006; Holtslag et al., 2013; Vignon et al., 2017).

We used the PALM (version 4.0, revision 1550) code in this study. The PALM code is developed by the PALM group at the Leibnitz University of Hannover, Germany (Maronga et al., 2015). This model solves the primitive hydro- and thermo-dynamic





equations for incompressible, Boussinesq fluid. We ran the model for dry atmospheric conditions. This choice simplifies the model setup and is motivated by the fact that the worst air quality was always observed during fair-weather conditions, under prolonged occurrence of temperature inversions in the Bergen valley. Thus, the model was initiated only with temperature and wind profiles. The initial wind profiles also serve as geostrophic wind profiles or forcing, which drives the PALM simulations

via the geostrophic wind term.

We ran PALM over a geographical domain centred on Bergen municipality. The domain spans 12.79 km in the zonal (East-West) and 17.27 km in the meridional (North-South) directions (left panel, Fig. 1). The surface geometry was set by the DEM-DSM data. The lateral boundary conditions in the model runs were periodic. The domain includes a 1000 m wide buffer zone at the outer boundaries of the model domain. This buffer was necessary to linearly interpolate the surface geometry, making it

periodic in both lateral directions. We set the model grid resolution to 10 m. The grid is vertically stretched by 1% for each subsequent grid level above 750 m. In total there were 128 vertical grid levels reaching 1450 m above the sea level, which is more than two times the height of the highest mountain peak within the model domain. The surface boundary conditions were different for the land and water surfaces. We used the Neumann (constant flux) condition for the land grid cells, as they have low heat capacity and quickly adjust the skin temperature. For the water grid cells, we used the Dirichlet (constant temperature)

condition, as water has a large heat capacity and retains an almost constant skin temperature over the simulation time window. Turbulent mixing is an irreversible process. It changes the temperature stratification and the wind profiles in the model also during the model spin-up. We counteract this difficulty by introducing a nudging routine. It relaxes the horizontally averaged temperature profile towards the initial temperature profile. This nudging is enabled only above the first local grid level over the surface. The relaxation time scale is $\tau = 43200$ s at elevations $z < 400$ m above the sea level. This time scale linearly

decreases to $\tau = 1800$ s at $z > 600$ m. Nudging is therefore very weak in the lower parts of the atmosphere, allowing the temperature profile inside the Bergen valley to be determined dynamically through the boundary conditions. In the free atmosphere above the mountains, we applied much stronger nudging, retaining the initial temperature profile. A simpler setup of the PALM model has been tested successfully in its application to the stably stratified ABL inside the Bergen valley (Wolf-Grosse et al., 2017a).

In order to simulate the dispersion of pollutants from the relevant sources, we used constant emission rates per grid cell. The emission rates were set separately for each of the three emission sources. Hence, $NO_2$ and $PM_{2.5}$ tracers from the harbour, chimneys and roads are simulated independently. In the applied model version of PALM, all pollutants are treated as passive tracers. We argue that this is a reasonable choice for the wintertime air quality assessment in a local high-latitude domain. The background near surface ozone ($O_3$) that could serve as a source of additional $NO_2$ from existing NO is greatly depleted during

the winter months. Sunlight that could lead to a photolytic conversion of $NO_2$ to NO and $O_3$ is largely absent. The nucleation of $NO_2$ into nitrate particles is slow compared to the transport and mixing processes. The particle growth beyond the $PM_{2.5}$ limits is also a slow process, whereas the particle wet scavenging is minimal under the clear sky conditions considered in this study. Gravitational settling of particles for the size range below 2.5 µm is slow.


## 3.2. Scenarios

The high-resolution environmental assessment and modelling is still a costly and computationally demanding exercise. Therefore, we selected the most relevant and impactful scenarios of high air pollution using joint statistical analysis of the long-term air quality and meteorological observations (Wolf-Grosse et al., 2017b; Wolf et al., 2014; Wolf and Esau, 2014).

Here we will not repeat the details of that analysis. Nevertheless, a brief summary might be useful for new readers. Studies of the extreme deterioration of the air quality may frequently benefit from the fact that the high concentrations of pollutants are reached after several hours (or even days) of persistently calm clear-sky weather. Such weather conditions are limited to a few specific sets of local values of the meteorological parameters. The high concentrations are mostly observed under south-easterly winds over the Bergen valley. Due to a complex interaction between the locally forced circulation and the large-scale

winds, the wind direction in the city is mostly south-easterly, too. That means, the air moves from the city towards the Bergen fjord. The local wind therefore transports air pollution from the locations with the most intensive emission and towards the fjord. The efficiency of this transport is dependent on a convergence zone that can, dependent on the interplay between the local and larger-scale drivers, be located either over the fjord or over the city areas in proximity to the harbour. Different inhabited areas might therefore be affected by emissions from some or all local emission sources.

### 3.2.1. Meteorological conditions

As the baseline scenario for this study, we used the dominant weather conditions, which were observed during the cases with measured hourly mean $NO_2$ concentrations above the legal threshold of 200 µg m$^{-3}$ at the DP site (Bergen Kommune, 2019). The initial and nudging temperature profile is the average of the temperature profiles measured over central Bergen during these high air pollution conditions as shown in Fig. 2. The geostrophic wind profile is the mean of the wind profile taken from

ERA-Interim over all high air pollution conditions. Due to the low resolution of ERA-Interim and the specific topography at the Norwegian west coast, the lowest grid level from ERA-Interim is located at around z = 410 m, dependent on large-scale air pressure. Therefore, the local scale flows are missing. To define a wind profile adapted to the realistic topography, we modified the wind speed profile to linearly increase from zero below z = 300 m to the actual ERA-Interim value at z = 450 m. Additional sensitivity experiments included the wind speed being 0.5 and 1.5 times of the baseline wind speed. The geostrophic

wind direction in the baseline scenario was set to wd = 110°. Sensitivity experiments used the wind directions of wd = 90° and wd = 130°, which cover the interval of the relevant wind directions found in the ERA-Interim data. The wind profiles for the different scenarios are shown in Fig. 3.

During each single PALM simulation, we kept the model forcing constant, meaning that we conducted simulations with static boundary conditions. This is a necessary simplification in order to reduce the computational efforts. Each scenario was

initialised with a precursor simulation running over 12 hours in order to stabilise the model circulation. At the end of the precursor run, the mean meteorological parameters were not drifting any longer. After the initialisation, the simulations were



continued for another six hours with emissions applied. For multiple simulations of the same meteorological condition, we used a restart option.

We do not have consistent surface heat budget observations in Bergen. To circumvent this problem, we inferred an approximate budget. The reviewed literature (Brümmer and Schultze, 2015; Nordbo et al., 2012) suggested as a reasonable value for the scenario a constant kinematic heat flux of $H_s$ = -0.02 K m s$^{-1}$ ($\approx$-20 W m$^{-2}$) over the land cells (lakes and sea inlets), we applied a constant surface temperature to reflect the very high heat capacity of the water mixed layer. We run three sensitivity experiments with the water surface temperature set to 0°C, 2.5°C, and 5°C. The baseline scenario is characterised by temperature inversions in the shallow ABL over Bergen. All meteorological conditions for the different scenarios are summarised in Table 2.

### 3.2.2 Emissions

We obtained the road traffic emission rates from the annual mean daily traffic data. For this, we converted the traffic counts in the grid cell $i$ of the PALM model to the emission rates, $R_i$, as

$$R_i = e_r \cdot N_i \cdot l_i$$

where $N_i$ is the annual mean daily number of vehicles (ADT) passing a certain grid cell; $l_i$ is the length of the road links per grid cell; $e_r$ is a suitable emission factor for the existing park of vehicles and their emission characteristics in Bergen. The calculation of the emission factors for $NO_2$ and $PM_{2.5}$ is described in detail in Appendix A.

We obtained the emission rates from wood-burning fireplaces (only $PM_{2.5}$), $C_i$, in a grid cell $i$ as

$$C_i = e_c \cdot n_i$$

where $n_i$ is the number of real estate properties with at least one registered wood-burning fireplace, per model grid cell $i$; $e_c$ is the typical emission factor for the wood-burning fireplaces. The calculation of this emission factor is described in detail in Appendix B.

Both emission rates $R_i$ and $C_i$ are defined as the surface fluxes in the model. It should be noted here that the *a posteriory* evaluation of the $PM_{2.5}$ concentrations in the simulations revealed that the provided emission rates from wood-burning fireplaces are too high. Therefore, we keep the concentration patterns but uniformly scale the calculated magnitudes by a factor of 0.1 in order to achieve reasonable concentrations. This inconsistency in the emission rates may be because either the emission rates per oven were overestimated or the fireplaces are much less used than suggested by the local fire department. There is also a degree of uncertainty related to the actual effective emission height. In the areas with the highest densities of chimneys, they typically have their exhaust at the level of the third-fourth floors. The effect of the emission height in the complex settings in the Bergen area should be assessed in more detail in future studies. However, at least the relative distribution of the pollutants should be reasonably represented assuming that the usage of ovens is similar between different neighbourhoods.

We obtained emission factors $S_i$ for the grid cell $i$, in which the offshore supply vessels typically are docked as



$$S_i = e_s \cdot v_s$$

where $e_s$ is the average emission factor per fuel spent for two representative ships provided by a local ship owning company; and $v_s$ is the typical fuel usage for the ships while at berth. Both numbers are factors describing the emissions from the ship's secondary engines that are smaller than the main engines used for travel and provide the ships with power while at berth. The

calculation of this emission factor is described in detail in Appendix C.

To compare to the most severe air pollution conditions in Bergen, all emission factors are calculated for typical high emission times, meaning daytime traffic, active usage of ovens for heating and standard emissions at berth for a busy day in the harbour in terms of the number of ships at berth.

## 4. Results

### 4.1. Baseline scenario simulations

The scenario ws01_wd01_ft01 is the baseline scenario for the air quality simulations (see Table 2). This scenario describes the most typical meteorological conditions during episodes with high air pollution in Bergen. The baseline scenario maintains a low-level (surface) temperature inversion in the lowermost 100 m above sea level almost everywhere over land in the Bergen simulation domain. The weak winds and wind channelling by the relief could be noted over the lowermost 400 m. Figure 4

shows the relief and vertical temperature and wind profiles in two selected areas of interest and 5 selected sub-areas for more detailed analysis. Area (a) represents the harbour area at the city center, while area (b) represents an area with dense population and heavy traffic at the southwest from the city center. Area (a) is strongly affected by the sea-land temperature contrast. The enhanced mixing over the water surface dilutes the temperature inversion there. Area (b) includes the heavily trafficked road junctions at DP and around (the sub-area 4). It also includes the Bergen meteorological observatory with a large amount of

meteorological instrumentation collocated in sub-area 3.

The wind speed and directions in the baseline scenario clearly indicate a low-level air transport in the Bergen valley towards its opening into Byfjorden – the sea inlet where the harbour is located. This channelled local circulation is likely enhanced by the sea-land temperature contrast as it has been described already through statistical analysis in Wolf et al. (2014) and dynamical analysis in Wolf-Grosse et al. (2017a). The latter study also addressed the distinct rotation in the wind-direction

from down-valley near the ground to up-valley at around $z = 300$ m and down-valley again above that altitude. This rotation with altitude is believed to be caused by an interaction between the large-scale meteorological circulation, the local topographic steering and the local forcing through the land-sea temperature contrast.

The temperature inversion in the simulated baseline scenario was most pronounced in sub-area 4. We assume this sub-area to be the most representative of the interior valley. The other sub-areas show more or less pronounced effects of the cold air

passing over warm water bodies. The simulated inversion profiles show multiple inversion layers that are not resolved in the observations. The maximum heights of the inversion profiles in the simulations are somewhat shallower than what is typically



observed. A more thorough discussion of the simulated and observed inversion profiles in Bergen can be found in Wolf-Grosse et al. (2017).

Due to its high resolution, the PALM simulations created a detailed geographical dispersion pattern of the $NO_2$ and $PM_{2.5}$ concentrations in and around the city. Figure 5 shows the $NO_2$ concentration pattern created by the road traffic emission in the

baseline scenario. The underlying geographical map (land and water surfaces) is given by grey shading as a Google Maps® picture. The use of geo-information tools to visualize the model simulations may facilitate the use of the simulated quantitative information in decision-making processes. The familiar map design simplifies orientation of and identification with the complicated pollution pattern.

The simulations revealed that emissions from road traffic are the dominant emission source for $NO_2$ over the populated parts

of the Bergen valley during high air pollution episodes. This was also confirmed with a detailed analysis of local pollutant measurements at the two reference stations (not shown). The baseline scenario indicates that the $NO_2$ concentrations are rather high (> 100 µg/m³) not only in direct proximity to the major roads but also in many adjacent urban areas. The exact shape and structure of the valley topography imprints strongly on the local dispersion conditions. Downwind of the major road transecting through the valley areas with a channelled flow appear. These are visible as streaks of elevated pollution concentrations

sometimes as high as 150 µg/m³. These streaks are separated by areas with pollutant concentrations below 50 µg/m³. The atmospheric channelling follows the areas with the lowest topographic height. In addition, the small water bodies (lakes and fjords) appear as areas with relatively lower air pollutant concentrations due to their comparatively higher surface temperature and the subsequent enhanced ventilation of the lowest air layers. This fine-scale structure of the dispersion pattern is especially relevant, since the simulations indicated elevated concentrations also in areas without continuous measurements with a

sufficient temporal resolution. Some urban areas might be affected by pollution transport over several kilometres and accumulation of emitted substances strongly different from the emission pattern.

The dominant emission sources for $PM_{2.5}$ during high air pollution episodes in the central Bergen valley turned out to be the wood-burning fireplaces. This was visible in the PALM simulations as well as in the detailed analysis of the available air pollution observations at the two reference stations (not shown). Therefore, one could expect the highest concentrations of the

$PM_{2.5}$ in the densely populated areas. Figure 6 shows the pattern of the $PM_{2.5}$ concentrations in the baseline scenario. Distinct to the $NO_2$ pattern, the $PM_{2.5}$ pattern is more evenly distributed over the entire city. The concentrations peak in the areas of the near-surface flow convergence in the lower parts of the relief. Since the wind is weak, areas with higher density of fireplaces are clearly identifiable (Fig. 1).

## 4.2. The simulated patterns from different emission sources

A major advantage of the high-resolution modelling is related to model's ability to simulate separately the impact and patterns of the different emission sources. As for $NO_2$, the most significant emission sources in Bergen are related to the ships in the harbour and the road traffic. This is likely a typical arrangement for many coastal cities around the globe. We have a reasonably good estimation of the absolute ship and traffic (road vehicles) emission rates and their spatial distribution to run independent





simulations of their pathways and the $NO_2$ concentration patterns. The combined assessment of the absolute and relative contributions from these two local air pollution sources is provided in Fig. 7. The road traffic emission dominates the urban air pollution almost everywhere in the city. The absolute emission rate per ship in port is, however, quite high. Assuming that within the assessed time, each vehicle is moved by 10 km, the emission from each ship would correspond to the emission from 1377 typical vehicles in Bergen. Using the concrete counts of the cars passing the central area, it gives us that 16 ships at berth in the harbour emit about 127% of the total car emission in the considered area. Such a significantly higher emission has, however, smaller contribution to the street-level concentrations of air pollutants. The ships emit at higher elevations. As the vertical mixing is strongly reduced, their emission does not reach the ground before they are either transported offshore over the Byfjord or diluted by the horizontal air movements.

There are three major sources of the $PM_{2.5}$ emission in Bergen, namely: the ships; the road traffic; and the wood-burning fireplaces. Figure 8 shows that the last source (fireplaces) absolutely dominates, even after its rescaling to provide more reasonable concentrations. The emissions per ship are approximately 34 times of that from a single wood-burning fireplace after applying the scaling factor of 0.1. However, the fireplaces are emitting at lower heights above the ground and clustered in the most populated area. The ships emit the $PM_{2.5}$ in the harbour area where the emitted pollution is, in many cases, effectively transported offshore and diluted over the unpopulated fjord area. The $PM_{2.5}$ concentrations from the road traffic are overall low.

At this point, however, the high uncertainty of $PM_{2.5}$ emissions should be emphasized, as emissions by road, tire and break abrasion have been neglected in this study in addition to the necessary correction of the emissions. Deposition of small and larger sized particulate (PM) may in addition play a crucial role and correct the simulated pattern to some degree. PM deposited on the ground can lead to high peak PM concentrations, when moist urban surfaces are drying off after several days with fair weather. This is especially relevant for major roads, where car induced turbulence can lead to a resuspension of dust greater than what the local wind conditions would allow for.

### 4.3. Air pollution pattern sensitivity to meteorological scenarios

The baseline weather scenario represents the most typical meteorological conditions leading to build-up of the air pollution in the city. The concrete observed weather conditions, however, vary and may differ from the considered scenario. Therefore, the air quality assessment needs to characterize sensitivity of the pollution distribution patterns to imposed perturbations of the meteorological parameters. The strong topographic steering in the valley restricts deviations in the simulated patterns produced by the sensitivity runs. Overall, the precursor sensitivity simulations showed very similar geographical concentration patterns but with varying accumulation strengths. The sensitivity scenarios are listed in Table 1. Only the scenarios with the most notable differences in the local dispersion conditions will be discussed below for shortness of presentation.

The stronger off-shore (easterly) wind (scenarios ws03_wd02_ft01 and ws03_wd02_ft02) deflect the pollutant plumes over the fjord (Fig. 9 for $NO_2$ and Fig. 10 for $PM_{2.5}$) reducing the pollutant transport out of the city centre and the harbour area. The waterfront takes most of the impact. The concentration patterns in the valley interior remain largely unchanged. We note that



the pollution patterns became more fragmented in those scenarios indicating somewhat reduced stagnation and accumulation of pollutants in the valley.

The most influential weather scenario with respect to the local dispersion conditions is ws03_wd02_ft03 (see Figs 11 and 12). A reversal of the flow at some elevation can be recognised from the up-valley transport of the emissions from ships in the

harbour in Fig. 11, while the wind vectors at the lower (z = 55 m) level still suggests a down-valley flow closer to the ground. This reversal was already visible in the vertical profiles in Fig. 4 but at the higher levels, so that it was invisible in the plume-dispersion at the surface (Fig. 7). The flow reversal is also vertically more extended in this simulation (not shown). The reason for this is a change of the interplay between the locally forced circulation due to the land-sea temperature contrast, the topographic steering and the large-scale meteorological circulation. The weaker convergence over the fjord due to the lower

fjord surface temperatures and the stronger large-scale winds weaken the down-valley circulation, reducing venting of the pollutants from inhabited areas to the water surface in Byfjorden. Hence, the populated areas close to the water front experience increased accumulation of pollutants, especially visible for the $PM_{2.5}$ concentrations in Fig. 12. This result is counter-intuitive, as a stronger large-scale wind allows for the increased accumulation. At the same time, the highest concentrations of $NO_2$ are reduced as the large road junctions in the interior of the valley are now affected by stronger winds and hence dispersion of the

pollutants.

## 5. Discussion

The routine monitoring of urban air quality with a few accredited measurement stations is nowadays the main instrument of environmental protection and control mechanisms. The decision-making process is frequently assisted with statistical assessment, zonation and forecasting of the concentrations of pollution levels. As the monitoring becomes cheaper, a more

detailed pattern of the urban concentration levels is gradually emerging (see e.g., the CurieuzeNeuzen project; CurieuzeNeuzen, 2019). For example, the area of the Bergen municipality is covered today with five stations measuring the air quality that characterize the concentrations in the most populous districts. However, as we demonstrate in this study, such an observational network is sub-optimal and not fully representing the complexity caused by the topography of a coastal valley environment. The typical spatial scales of the surface topography and heterogeneity require in principle a denser monitoring

network and refinement of the forecast models in order to reach a deeper understanding of the local dispersion conditions at adequate spatial scales. In the case of Bergen, the model's resolutions of a hundred meters or finer is necessary in order to resolve the processes for distribution and accumulation of air pollutants near the surface.

Such high-resolution models are now available. The existing computer capacity allows running the model simulations for several typical meteorological scenarios, which are associated with the typically observed air pollution episodes. In Bergen, it

was found that only a few of the meteorological scenarios result in high pollution concentrations (Wolf et al., 2014). Therefore, the high-resolution scenario modelling is not only technically feasible but also would create an added value for the risk and vulnerability assessment for urban areas. In future, high-resolution models could be coupled with existing routine numerical


weather forecasts and data assimilated from measurement stations. The present study was, however, limited to the air quality assessment tasks, which are related to a local policy making process. It does not intend to advance to the routine local air quality forecast.

All emissions of air pollution is harmful to the environment, however in development of pollution policies and mitigation strategies quantification of the major sources of pollution and in particular their impact on the observed elevated concentrations of the pollutive substances at the street level is needed. This is a difficult assessment task as the emitted pollution is transported and dispersed by an intricate pattern of the turbulent local flows in an urban environment. The dynamic nature of the concentration patterns and their sensitivity to the variations of the meteorological conditions are not fully considered. This study demonstrates that the use of high-resolution models can contribute to overcome this challenge. The models can simulate the dynamics and specific contribution of each of the individual pollution sources into the total concentration of each pollution category. The simulation results might be utilized to tune policies in a dialog with the major polluters. In the presented case of the Bergen harbour authorities, the harbour administration assesses the simulated concentration patterns to minimize the impact of the ship exhaust on the city air quality.

It should be noted here that the high-resolution dispersion pattern produced in this study is based on physical meteorological fields with an equally high resolution, thus including the effect of topographic steering and other local forcing, consistent with how the simulations are defined. Other attempts to produce high resolution dispersion patterns in Bergen and other places are based on lower resolution meteorological fields but high-resolution emission maps (e.g. Miljødirektoratet, 2019). This gives the maps a faulty high-resolution appearance that however neglects the important impact of local flow steering.

The high-resolution modelling is a relatively new approach in meteorology and air quality science. Several scientific challenges need to be resolved in future studies. Model simulations with this amount of spatial detail are difficult to validate against observations. While it is possible to state that the simulated pollutant pattern and meteorological conditions are reasonably similar to the observations, a rigorous validation is challenging. Usual validation against operational weather stations run by meteorological services is not possible due to a too low spatial density of the observations. Including the abundantly available citizen observations might be a pathway for future validation (Johansson et al., 2015; Schneider et al., 2017; Zilitinkevich et al., 2015) but goes beyond the scope of this study. General validation of the modelling technique is required with dedicated measurement campaigns or experiments (e.g. Hertwig, 2013). The Bergen test bed with its high density of meteorological observations could serve as a validation case.

Initialisation of the model simulations should be improved in order to increase correspondence between observed and modelled conditions (e.g. Maronga et al., 2019). This is necessary both, for improving the realism of the simulations but also for their validation. The subset of simulated conditions needs to be compared to the relevant subset of observed conditions. For this, a better correspondence with observed cases is necessary. It is clear that model simulations with periodic boundary conditions, as it has been used in this study, are only an intermediate step. Simultaneously, a reduction of the computational costs should be assessed. Both could be achieved through nesting of model domains with different spatial resolutions as it is already routinely done for coarser scale simulations with e.g. the WRF model (Cécé et al., 2016; Muñoz-Esparza et al., 2017).





## 6. Conclusions

This study applied the turbulence-resolving model PALM in the local air quality context to assess the conditions and consequences of low-level emission of nitrogen dioxide ($NO_2$) and particulate matter ($PM_{2.5}$) from the major polluters in Bergen, Norway. We ran simulations at 10 m resolution over a very large geographic domain spanning 12.79 km in the zonal
(East-West) and 17.27 km in the meridional (North-South) directions. In order to assess the possible bandwidth of circulation conditions inside the valley we conducted a set of eight sensitivity runs with varying model forcing in addition to the baseline scenario reflecting the most typically observed air pollution conditions. We simulated separately the emissions from the three major local emission sources of air pollution. These are car traffic, heating in wood-burning fireplaces and emissions from offshore supply vessels docked in the harbour area, located next to the city centre.

The results support the following conclusions: The very high-resolution numerical simulations change our perspective on the magnitude and dynamics of the air quality under the most typical air quality hazard scenarios. Small topographic features like the shape of the valley floor or local water bodies strongly affect the dispersion and accumulation conditions. This results in areas with a channelled flow, where air with a high pollution load (in excess of 150 µg/m$^3$ for $NO_2$) can be transported over several hundred meters or even kilometres. Especially for $PM_{2.5}$ both, areas with maximum emissions, but also areas with
minimum ventilation are clearly visible in the dispersion maps. The sensitivity runs highlight the relevance of the interplay between the local surface conditions and the larger-scale circulation. Attempts to model the dispersion of pollutants in this city at resolutions that are unable to resolve this interplay and the local fine-scale topographic features, will most likely fail to produce the necessary details of the dispersion map.

The separate analysis of emissions from the three major emission sources was helpful in a source-appointment of the overall
pollution levels in the city centre. An assumed high impact of the ships located in Bergen harbour as the major polluter in the city was not generally confirmed in the simulations. Despite the 16 ships in the harbour emitting 27% more $NO_2$ than all cars in the simulated domain, their relative impact on the air pollution over inhabited parts of the valley exceeds 25% over a larger area only for one of the simulated scenarios. This could however be different for other, similar harbours, as our simulations attributed this lower impact to the intrinsic structure of the local circulations at the coast.

The usage of high resolution data and numerical model simulations for meteorological services gives a so-far unprecedented amount of detail for end-users and allows for a direct connection of the scientific issues with an understanding of the societal dimensions as stated in the PEEX white paper (Lappalainen et al., 2016). The Bergen Harbour Authority (BOH) deemed the results from this study as helpful for their efforts to reduce the impact of emissions from their ships on the local population. This study serves as a demonstration of a concept for an approach of statistical dynamical downscaling applied to high
resolution services by making LES usable for a reduced and therewith feasible amount of possible model simulations under selected meteorological scenarios.



**Author contribution**

All co-authors conceptualised the study and designed the applied methodology. LHP and TW collected the data. LHP and TW acquired the funding. IE was the project manager. IE and TW designed and applied the software. IE had a role of supervision for TW. TW developed the visualisation with support from IE and LHP. IE initialised the manuscript. TW wrote the
5    manuscript. All authors contributed to the final review and editing of the manuscript.

**Acknowledgements**

This study used external data from the following sources: Endre Leivestad and Trond Grindheim - Bergen Municipality, Haso Bradaric – Norwegian Mapping Authority, Ole Edvard Grov and Joachim Reuder – University of Bergen, Rita Våler – Norwegian Institute for Air Research, Stig Nyland Andersen – Norwegian Public Road Authority, Even Husby – BOH, and
10   the European Centre for Medium-Range Weather Forecasts (ECMWF). The PALM model is developed and maintained at the Institute of Meteorology and Climatology, Leibniz University Hannover. CPU time was provided through the Norwegian Supercomputing Project (NOTUR II grant numbers nn2993k and nn9528k). This study was funded by Bergen Harbour Authority (BOH). Ulrik Jørgensen, Sverre Østvold, Nils Møllerup and Even Husby - BOH, and Eva Britt Isager and Per Vikse – Climate section, Mette Iversen and Nils-Eino Langhelle – Section for Plan- and Geodata, Per Hallstein Fauske and Arve
15   Bang, Health Care Agency, all Bergen municipality, provided the user-perspective and valuable cooperation. Initial developments were funded through the GC Rieber foundation PhD fellowship.

The illustrations were prepared as a colour-coded semi-transparent contour and shading layers on top of a Google Maps background (http://code.google.com/apis/maps/, using the get google map version 2.0 function from the MathWorks file exchange https://se.mathworks.com/matlabcentral/fileexchange/27627-zoharby-plot-google-map).



**Table 1. Data sets used for this study**

| Dataset | source | Format, data type (period) |
|---|---|---|
| **Topographic data** | | |
| Laser measurements from over the square area of size 10 x 10 km centred over Bergen city hall. | Bergen municipality, already combined and filtered in previous project | LAS point cloud |
| Topographic height for Bergen municipality | Bergen municipality | DSM, GeoTiff, 10 m horizontal resolution |
| Topographic height for surrounding municipalities | Norwegian mapping authority | DSM, GeoTiff, 10 m horizontal resolution |
| Water surfaces | Bergen municipality | Shape file, polygon |
| **Local measurements and large-scale circulation** | | |
| Microwave radiometer measurements (inversions) | Own data, NERSC | ASCII, time series (2011-2016) |
| Weather stations Geophysical Institute | University of Bergen (UiB) | ASCII, time series (2011-2016) |
| ERA-Interim (large scale meteorology) | European Centre for Medium Range Weather Forecasts (ECMWF) | Netcdf, map with different resolutions and time scales |
| Weather stations at Skolten and Jekteviken | BOH | ASCII, time series (2014-2016) |
| Air pollution measurements | Norwegian Institute for Air Research (NILU, 2019) | ASCII, time series (2003-2016) |
| **Emission data** | | |
| Main streets (centre line and traffic) | Bergen municipality | Shape file, traffic information |
| Side-streets (centreline and traffic) | National road authority | Shape file, traffic information |
| Traffic at Danmarksplass | National road authority | Excel list with traffic counts for different time periods |
| Properties with fire places | Bergen fire department | Shape file, point data, list |
| Harbour log | BOH | Excel list with ship type, name, arrival and departure, docking time (2015-2016) |



**Table 2: Parameters for the different meteorological scenarios. \***

| Parameter | Name | Description |
|---|---|---|
| Wind-speed | ws01 | Vertical profile baseline scenario |
| | ws02 | 0.5 times baseline scenario |
| | ws03 | 1.5 times baseline scenario |
| Wind-direction | wd01 | Baseline scenario, 110° |
| | wd02 | 90° |
| | wd03 | 130° |
| Surface temperature Bergen fjord | ft01 | Baseline scenario, 2.5°C |
| | ft02 | 5°C |
| | ft03 | 0°C |

**\*The syntax for the identification of model scenarios is boh\_ followed by the different meteorological parameters (e.g. boh\_ws01\_wd01\_ft01 for the baseline scenario).**

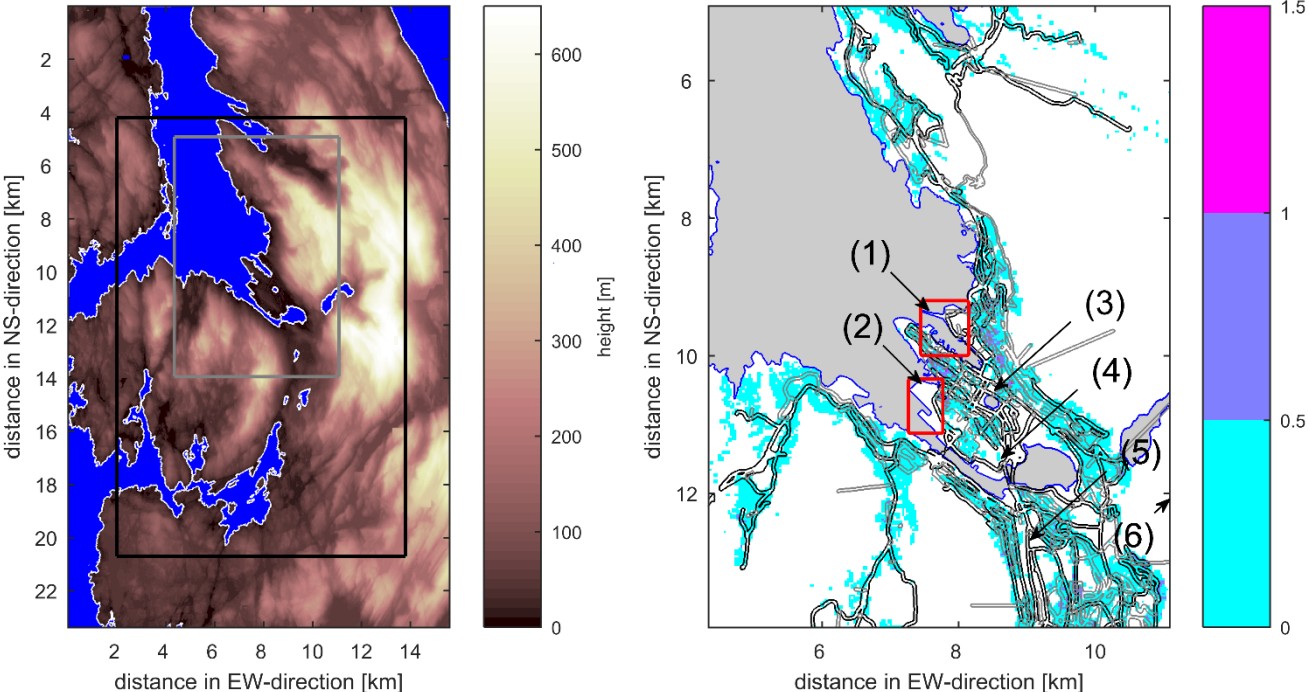

**Figure 1: Left: Topographic map of the Bergen area. The black square indicates the final model domain used for the PALM simulations; the grey square indicates the focus area for the analysis of the PALM simulations. Right: Emission map for Bergen City Centre used in the PALM simulations. Colour shading indicates the number of parcels of land with registered active fireplaces (for domestic heating) per 10 x 10 m² grid-square. Black/grey lines indicate the location of main/side roads. The main harbour areas Jekteviken (southwest) and Skolten (northeast) are indicated with two red squares. The numbered arrows point at the location of the automatic weather stations at (1) Skolten, (2) Jekteviken, (4) Florida and (6) Ulriken, and the air pollution stations at (3) Rådhuset and (5) Danmarksplass.**





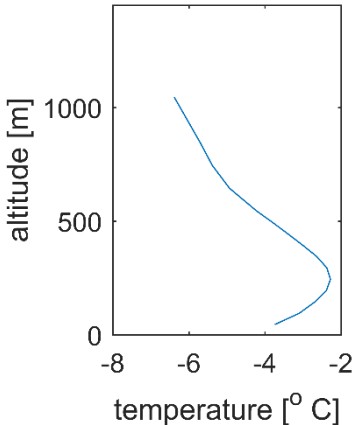

**Figure 2: The mean vertical temperature profile used for nudging in the PALM domain.**





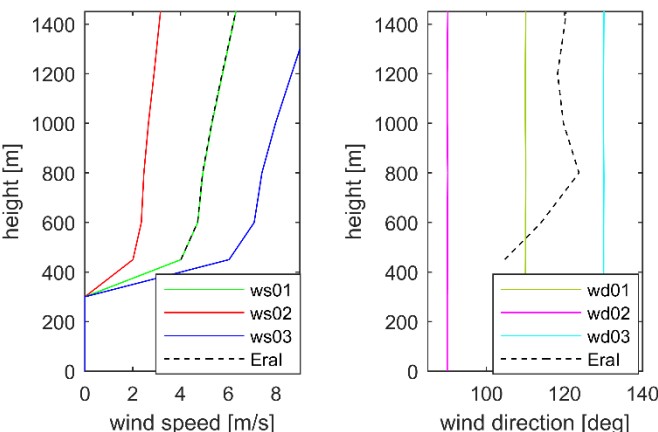

*Figure 3: Mean wind speed (left panel) and wind direction (right panel) profiles in the considered scenarios and sensitivity experiments.*



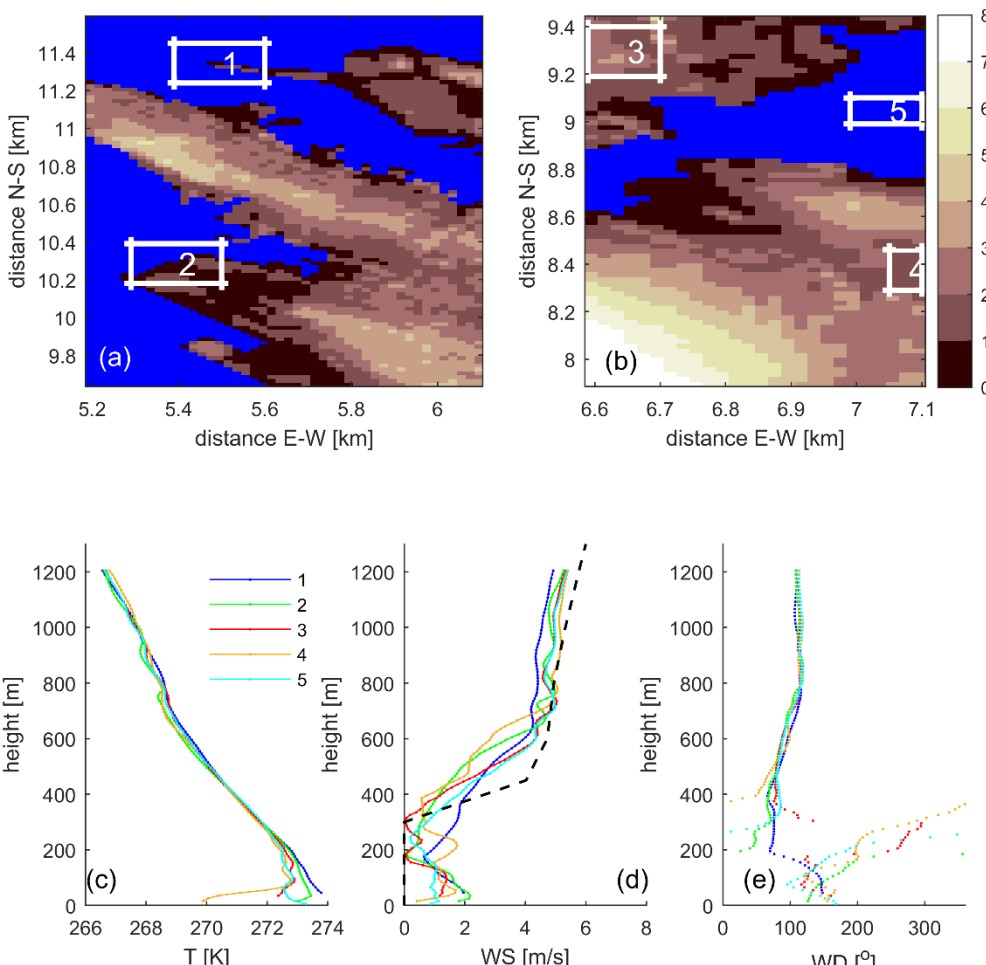

**Figure 4: The meteorological conditions averaged over the last 15 min of the 12 h precursor simulation for the baseline scenario ws01_wd01_ft01. The conditions were sampled in two areas: (a) the harbour area at the northwest end of the city centre where the profiles 1 and 2 were sampled; (b) the heavy traffic and densely populated area at the southeast end of the city centre where the profiles 3, 4 and 5 were sampled. The temperature profiles (c) and the wind speed (d) and direction (e) profiles were averaged over the sub-areas shown by the white rectangles.**

**Figure 5: The simulated NO₂ concentration pattern in the baseline scenario ws01_wd01_ft01 in the central part of the model domain. The emission source is the road traffic (cars). The concentrations are given by semi-transparent colour shading. The concentrations were sampled at 5 m above the surface. Concentrations below 5 µg/m3 are omitted. The wind vectors characterize the flow 55 m above the surface. The sampled data were averaged over the last 15 min of the 6 h dispersion run. The underlaying grayscale land and water surface was taken from Map data @2019 © Google.**



**Figure 6: The same as in Fig. 5 but for the PM₂.₅ concentration pattern. The emission source of PM₂.₅ here was associated with the wood-burning fireplaces. The underlaying grayscale land and water surface was taken from Map data @2019 © Google.**



**Figure 7: The same as in Fig. 5 but for the road traffic emission (red shading) and the ships in the harbour emission (green shading) plotted together through the artificial colour palette. Concentrations below 5 µg/m3 are omitted. The relative contribution of the ship emissions into the total local NO₂ concentration are given by contours. The underlaying grayscale land and water surface was taken from Map data @2019 © Google.**



**Figure 8:** **The same as in Fig. 7 but for the PM$_{2.5}$ emissions from the road traffic (red shading), ships in the harbour (green shading) and wood-burning fireplaces (blue shading). Due to the dominating impact of the emissions from wood-burning fireplaces, the red and green colours are only visible at their emission hot spots in the harbour and at the major roads in the south-eastern part of the domain. The underlaying grayscale land and water surface was taken from Map data @2019 © Google.**



**Figure 9: The same as in Fig. 7 but for the scenario ws03_wd02_ft02. The underlaying grayscale land and water surface was taken from Map data @2019 © Google.**



**Figure 10. The same as in Fig. 8 but for the scenario ws03_wd02_ft02. The underlaying grayscale land and water surface was taken from Map data @2019 © Google.**





**Figure 11. The same as in Fig. 7 but for the scenario ws03_wd02_ft03. The underlaying grayscale land and water surface was taken from Map data @2019 © Google.**



**Figure 12. The same as in Fig. 8 but for the scenario ws03_wd02_ft03. The underlaying grayscale land and water surface was taken from Map data @2019 © Google.**





**Appendix A: Emissions from road-traffic**

The emissions from road-traffic vary strongly with the composition of the vehicle fleet passing a road link. We calculated emission factors specifically for the vehicle fleet at use in Bergen.

Traffic flow information from the national road authority overlaps with the traffic flow information from Bergen municipality
(section 2.2). Both include information on main roads. We assume the measurement-based traffic flow information from Bergen municipality to be more representative than the model-based traffic flow information from the national road authority. We therefore combined both into one consistent dataset giving preference to information from Bergen municipality wherever possible. The information used from the two datasets are shown in Fig. A1.

We decided to neglect the emissions from tunnels. This is in contrast to another study recently conducted for the Bergen area
based on statistical modelling (Denby, 2014). This study split all emissions from within tunnels equally to both ends. This approach neglects the ventilation system especially at use in longer tunnels that always runs either towards one side or removes air through vertical ventilation shafts at some distance from the tunnel ends, mostly away from inhabited areas. This will lead to an underestimation of street emissions at the positions of the ventilation shafts and some tunnel exits but avoids the problem of an unknown overestimation of emissions at all tunnel openings.

For the separation of the vehicle fleet in Bergen into different vehicle classes we used information from an Oslo-based survey (Hagman and Amundsen, 2011). In this survey small passenger cars account for 72 % of all road-traffic. The remaining 28 % are composed of small diesel driven delivery cars (15 %), heavy transportation with lorries (10 %) and busses (3 %). The passenger cars are furthermore divided into 47.6 % diesel driven cars, 49.7 % gasoline driven cars and 2.6 % electric cars. Due to the high road taxes for any other than electric cars and the increasing number of electric cars over time, especially in the
Bergen area, we assume that electric cars are used four times as often than other types. This leads to the final distribution of passenger cars of 44 % diesel driven, 46 % gasoline driven and 10 % electric. From the 3 % bus traffic, 27 % are gas driven (Målfrid Vik Sønstabø, public transport authority, personal communication, 2016). These are assumed to have negligible emissions of $NO_2$ and $PM_{2.5}$.

For the specific emissions per driving distance we use constant values for all streets in Bergen and a constant composition of
the traffic pattern. Due to a lack of information on the specific vehicles, speeds and driving patterns along each street we assume all vehicles to have emission class Euro 5 and that the vehicles are standing in congested traffic for 50 % of the time. This gives some errors due to different driving pattern or vehicle-fleet compositions along specific roads, but a more detailed specification of the emissions would be beyond the scope of this work. The emission factors per distance and vehicle type are stated in Table A1.

The traffic flow information is given in annual mean daily traffic flow. Typically, the highest $NO_2$ pollutant concentrations are measured between 07:00 UTC and 17:00 UTC during work days and in February. For the assessment of the traffic during these times we used the traffic information from Danmarksplass, close to the DP air pollution reference station. This area should be representative of the traffic in Bergen, as it is one of the main streets handling the city traffic. During the 10 hours 67 % and





75 % of all traffic happens with passenger cars/ small delivery vans and heavy lorries/city buses, respectively. In addition, 1.125 and 1.011 times of the annual mean daily traffic happens during work days and February for passenger cars/small delivery vans. The corresponding numbers for heavy lorries/ city buses are 0.965 and 1.298, respectively.

The resulting total emission factors for road-traffic are thus, $e_r = 5.54 \cdot 10^{-5}$ µg/(m$^3$ · s) for NO$_2$ and $e_r = 2.10 \cdot$

5   $10^{-6}$ µg/(m$^3$ · s) for PM$_{2.5}$ distributed over the 100 m$^2$ large grid cells. All emissions from road traffic are averaged over 3 x 3 grid cells to follow the width of large streets, where most of the emissions take place and include the effects of vehicle-induced turbulence.



**Tabell A1: Emission factors $e_r$ for NO₂ og PM₂.₅ for the different vehicle types.**

| Vehicle type | NO₂ (g/km) * | | Mean emission factors (µg/m³s) ** | PM₂.₅ (g/km) * | | Mean emission factor (µg/m³s) ** |
|---|---|---|---|---|---|---|
| | Congested city traffic | Freely flowing city traffic | | Congested city traffic | Freely flowing city traffic | |
| Passenger car diesel | 0.350 | 0.183 | | 0.0043 | 0.0017 | |
| Passenger car gasoline | 0.004 | 0.002 | 2.51 10⁻⁵ | 0.0009 | 0.0006 | 3.52 10⁻⁷ |
| Passenger car electric | 0 | 0 | | 0 | 0 | |
| Small delivery vans | 0.300 | 0.217 | 5.47 10⁻⁵ | 0.0050 | 0.0021 | 7.51 10⁻⁷ |
| Heavy lorries | 0.899 | 0.481 | 1.79 10⁻⁴ | 0.0904 | 0.0385 | 1.68 10⁻⁵ |
| City bus diesel | 2.530 | 1.430 | 3.76 10⁻⁴ | 0.0083 | 0.0041 | 1.22 10⁻⁶ |
| City bus gas | 0 | 0 | | 0 | 0 | |

**\* Source: Hagman and Amundsen (2011).    \*\*Calculated for this study.**





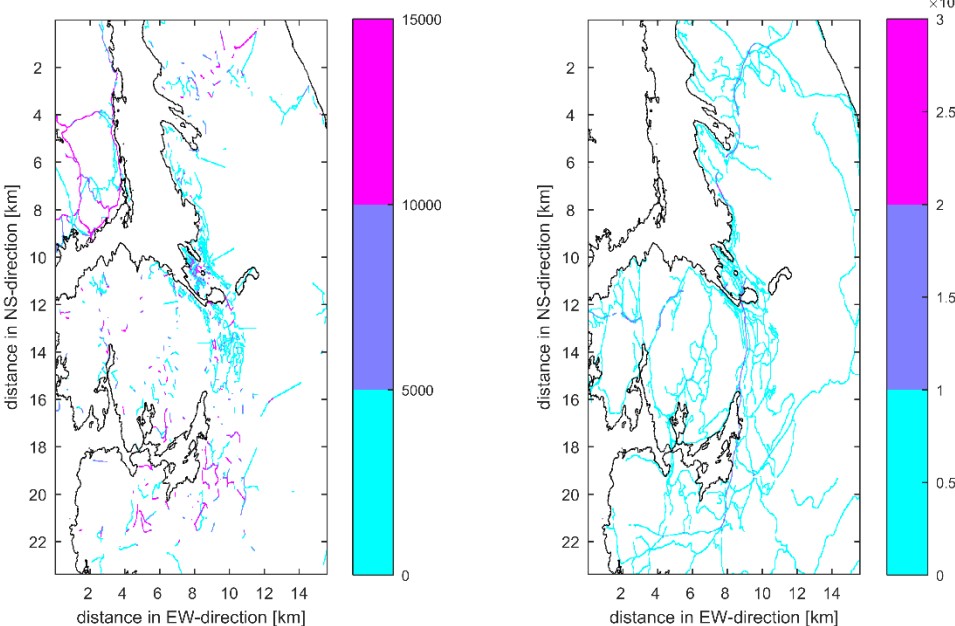

**Figure A1: Annual mean daily traffic flow in vehicle meters ($N_i \cdot l_i$) in each PALM grid cell for side-streets (left panel) and main roads (right panel). Both colour shadings have the unit [m]. Roads in the left panel that are appearing to go straight into mountainous areas are sporadically used small utility roads. For simplicity of the underlying traffic model they are indicated as straight roads instead of following their actual path. Emissions from these roads are negligible.**



## Appendix B: Emissions from wood-burning fireplaces

The emission factors from wood-burning fireplaces assume constant emissions per oven according to the existing mix of oven types in Bergen and their estimated typical usage (both provided by the Bergen fire department). According to these numbers, there are 25 % modern ovens with catalysator, 25 % modern ovens without catalysator and 50 % old ovens presently in use in

5    Bergen. These have emissions factors of 5, 10 and 30 g/kg burned wood, of which 90 % are assumed in the size fraction $PM_{2.5}$. Like the street related emissions, we averaged the emissions per oven over 3 x 3 grid cells.

70 % of all wood-burning fireplaces are assumed in use during cold winter days. In addition, each property that is registered to have fireplaces might have at least one fireplace, possibly more. Together it is therefore roughly assumed that one wood-burning fireplace per property might be used on a daily basis. Figure A2 shows the number of wood-burning fireplaces per

10    grid cell after special averaging. During a typical firing cycle, a total of 11 kg wood are assumed to be used and we assume that most firing takes place between 16:00 and 23:00 UTC, after people come home from work. The final emission factor per property with a fire place registered is therefore $e_c = 74.21$ $\mu g/m^2 s$ for $PM_{2.5}$ distributed over the 100 $m^2$ large grid cells. wood-burning fireplaces are assumed not to contribute significantly to the local $NO_2$ emissions.





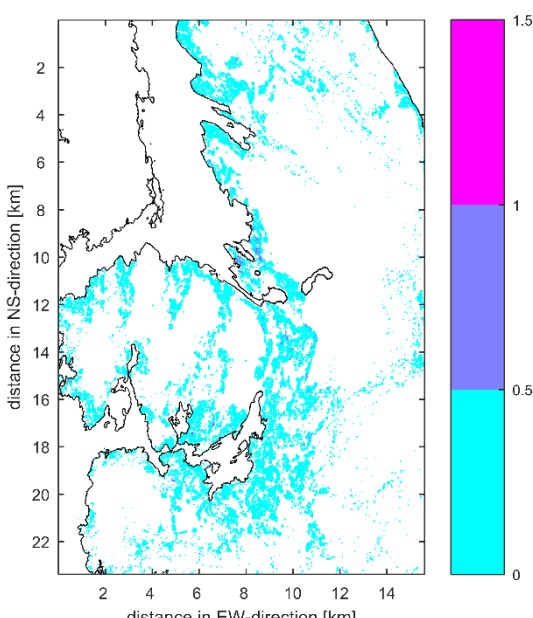

**Figure A2: Number of wood-burning fireplaces per PALM grid cell ($n_i$) after spatial averaging**




## Appendix C: Emissions from ships

Figure A3 shows the position of the harbour docks frequented by supply ships in Bergen together with the number of hours with supply ships docked at these locations from January 2015 to March 2016. In order to identify the most typically used locations, we specified emissions in PALM from all harbour docks with more than 2000 h of ship docking time in the analysed period. This reduced the number of supply ships with emissions specified in the PALM simulations to 16. This can be seen as a higher threshold for the number of supply ships typically in the harbour, describing a busy day. For the emissions we assumed that the ships were running their secondary engines for on-board energy supply during docking.

For the assessment of the emissions per ship we used the mean numbers for two offshore oil-exploration related ship types that are the most typical in Bergen. These are the smaller offshore platform supply vessels (represented by Normand Carrier, Solstad Shipping) and the larger anchor handling vessels (represented by Normand Ranger, Solstad Shipping), both are for simplicity reasons referred to as offshore supply vessels in this study. The larger ship had a weight of 3051 gross tonnage (GT) and typical $NO_x$ emissions of 41.05 g/kg spent fuel. The smaller ship had a weight of 4750 GT and typical $NO_x$ emissions of 39.28 g/kg spent fuel. According to Volker Matthias (Helmholtz Centre for Materials and Coastal Research Geesthacht, personal communication, 2016) of the total $NO_x$ emissions from these ships 10 % are $NO_2$. The average $NO_2$ emission of both ships is therefore $e_s = 4.02 g/kg$ spent fuel. We calculated the typical fuel usage of the ships based on their weight as suggested in Hulskotte et al. (2014). The typical mean fuel usage for both ships, when at berth for these two ships were $v_s = 67$ kg/h for the mean ship weight of 3900 GT. This results in a typical emission factor per supply ship of 275 g/h or $S_i = 763.31$ µg/(m$^2 \cdot$ s) distributed over the 100 m$^2$ large grid cells. For the PM$_{2.5}$ emissions we used $e_s = 1.5$ g/kg spent fuel for both ships with 90 % of the particles in the PM$_{2.5}$ size range (Volker Matthias, Helmholtz Centre for Materials and Coastal Research Geesthacht, personal communication, 2016). This led to emission factors of 90 g/h or $S_i = 251.03$ µg/(m$^2 \cdot$ s) distributed over the 100 m$^2$ large grid cells for PM$_{2.5}$. All these numbers assume that no specific fuel cleaning technology was in use while the ships are at berth as recommended by Even Husby (BOH, personal communication, 2016). The emission numbers are in agreement with measurements of ship plumes some distance from the ships in Finland (Pirjola et al., 2014).

In order to account for the elevated emission from ships we artificially increased topography at the location of the ships to a typical height of the chimneys on top of the assessed supply vessels plus an offset to account for initial plume rise. This way we could specify the emissions as fluxes from the ground. Elevated fluxes were not readily available in the used PALM version. The total increase of the topographic height at the location of the ships is 40 m. This gives an elevated plume rise but not too strong disturbance of the local flow pattern, since the increase in topographic height for the entire domain only covered 16 grid points. Additional plume rise is considered by setting a positive surface heat flux on top of the elevated topography of 1000 W/m$^2$.




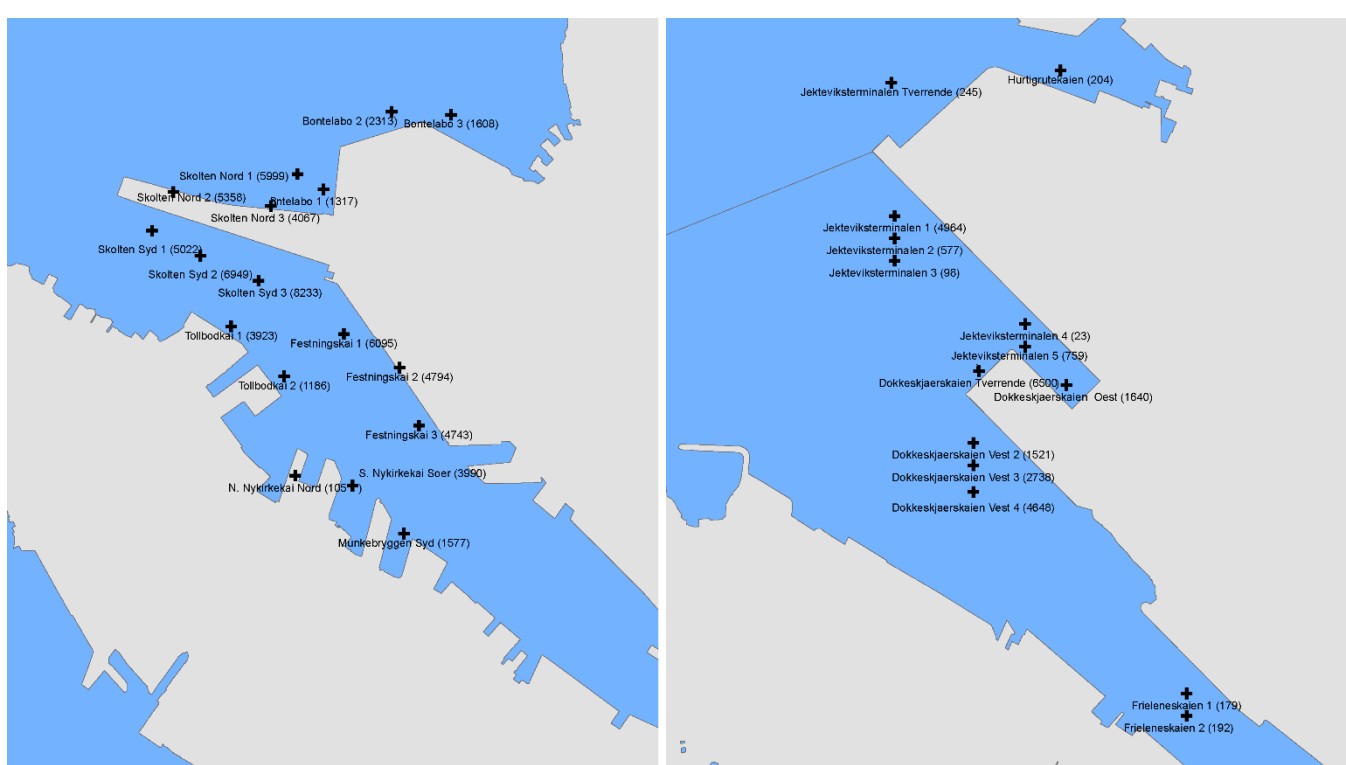

**Figure A3: Ship positions in Bergen harbour. The numbers behind the names of the harbour docks indicate the number of hours with supply ships docked at each of the harbour docks from January 2015 to March 2016.**





**Appendix D: A list of research peer-review articles and on-line materials certifying the quality and applicability of the PALM model for the urban air quality, turbulent dynamics and pollution diffusion studies.**

Castillo, M. C. L., Kanda, M. and Letzel, M. O.: Heat ventilation efficiency of urban surfaces using large-eddy simulation, Annu. J. Hydraul. Eng., 53, 175–180, 2009.

GRONEMEIER, T., INAGAKI, A., GRYSCHKA, M. and KANDA, M.: Large-Eddy Simulation of an Urban Canopy Using a Synthetic Turbulence Inflow Generation Method, J. Japan Soc. Civ. Eng. Ser. B1 (Hydraulic Eng., 71(4), I_43-I_48, doi:10.2208/jscejhe.71.i_43, 2016.

Kanda, M., Inagaki, A., Miyamoto, T., Gryschka, M. and Raasch, S.: A New Aerodynamic Parametrization for Real Urban Surfaces, Boundary-Layer Meteorol., 148(2), 357–377, doi:10.1007/s10546-013-9818-x, 2013.

Kurppa, M., Hellsten, A., Auvinen, M., Raasch, S., Vesala, T. and Järvi, L.: Ventilation and air quality in city blocks using large-eddy simulation-urban planning perspective, Atmosphere (Basel)., 9(2), 1–27, doi:10.3390/atmos9020065, 2018.

Letzel, M. O., Krane, M. and Raasch, S.: High resolution urban large-eddy simulation studies from street canyon to neighbourhood scale, Atmos. Environ., 42(38), 8770–8784, doi:10.1016/j.atmosenv.2008.08.001, 2008.

Letzel, M. O., Helmke, C., Ng, E., An, X., Lai, A. and Raasch, S.: LES case study on pedestrian level ventilation in two
neighbourhoods in Hong Kong, Meteorol. Zeitschrift, 21(6), 575–589, doi:10.1127/0941-2948/2012/0356, 2012.

Maronga, B., Gryschka, M., Heinze, R., Hoffmann, F., Kanani-Sühring, F., Keck, M., Ketelsen, K., Letzel, M. O., Sühring, M. and Raasch, S.: The Parallelized Large-Eddy Simulation Model (PALM) version 4.0 for atmospheric and oceanic flows: model formulation, recent developments, and future perspectives, Geosci. Model Dev., 8(8), 2515–2551, doi:10.5194/gmd-8-2515-2015, 2015.

Maronga, B., Gross, G., Raasch, S., Banzhaf, S., Forkel, R., Heldens, W., Kanani-Sühring, F., Matzarakis, A., Mauder, M., Pavlik, D., Pfafferott, J., Schubert, S., Seckmeyer, G., Sieker, H. and Winderlich, K.: Development of a new urban climate model based on the model PALM – Project overview, planned work, and first achievements, Meteorol. Zeitschrift, 28(2), 105–119, doi:10.1127/metz/2019/0909, 2019.

Park, S.-B., Baik, J.-J., Raasch, S. and Letzel, M. O.: A Large-Eddy Simulation Study of Thermal Effects on Turbulent Flow
and Dispersion in and above a Street Canyon., J. Appl. Meteorol. Climatol., 51(5), 829–841, doi:10.1175/JAMC-D-11-0180.1, 2012.

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
