# Peer review of "A very high-resolution assessment and modelling of urban air quality"

_Atmospheric Chemistry and Physics, 2019_

## Referee Comment (RC1) · Anonymous Referee #1 · 22 Aug 2019

**General Comments**

Within the manuscript, an example/template urban air quality assessment is presented for the city of Bergen, Norway. The air quality assessment is conducted by means of large-eddy simulations (LES) utilizing the LES model PALM. The presented work analyses $NO^2$ and PM2.5 concentrations emitted from different sources which are ships, cars and domestic fireplaces within the city and harbour of Bergen and the close surroundings. The work is aimed to give a first idea how such air-quality assessments can provide more detailed information about the impact of pollutant emissions on the urban air quality. The fact that the different pollutant sources are treated independently also allows for a better identification of the main source of pollution. This, in turn, helps decision-making authorities to better plan mittigation measures in order to ef-

fectively reduce pollution within the city. Results of this study are already used by the local harbour authority for reducing pollution caused by ships. The presentation of the possibility for such high-resolution air-quality assessments and their value to the local authorities and citizens, is of great value to the scientific community. Open research questions and missing input information for such applications are presented. Follow-up studies can benefit from these results and work on further improvements of air-quality assessments. I, therefore, suggest the manuscript for publication after the following recommended changes are implemented into the manuscript.

**Specific Comments**

Within the description of the data sets, it is mentioned that only supply vessels for the offshore oil industry are considered in the pollution emission of ships (p.5, l.9). This raises the question why not all ships are considered. There is no information given if these type of ships are the main pollutant emitter or if they contribute only by little to the total ship emissions. As the analysis of the contribution of ship emissions to the total air pollution is one of the major points of the manuscript, this question must clearly be answered within the text.

During the presentation of the results in Sect. 4, the different used scenarios are mentioned at various points. Although, in Sect. 3.2.1, all considered meteorological conditions are mentioned and listed in Table 2, a list is missing which shows the actually conducted simulations with the used combinations of parameters. Such a table is also referenced on page 11, line 29 but is not part of the manuscript. Also, the naming of the scenarios is not explained in the text. An explanation is given below Table 2, but the explanation is incorrect and needs to updated.

In the discussion section, it is mentioned, that a resolution of at least 100m should be used to correctly resolve the diffusion processes of air pollutants within the Bergen region (p.12, l.26). I am missing a justification for this statement. Why can I also use 100m for such a study? Is there a gird-sensitity study conducted which suggested a

resolution of 100m or finer? A quotation is needed at this point. Otherwise, this has to be corrected to 10m resolution as this is the resolution used for this study.

**Typos and other Technical Corrections**

On page 10, line 26, it is mentioned that areas with higher density of fireplaces can easily be identified using Fig. 6 and Fig. 1. In my opinion, it is very hard to compare Fig. 1 and Fig. 6 because I cannot identify the coastlines in Fig. 6 (and also in Figs. 5-12). As I am not familiar with the area around Bergen, I need some help to navigate within the figures and the coastlines would drastically help. My suggestion is, either try to draw the coastlines into Figs. 5-12 or remove the white colour from the colour shading.

- p.4, l.9: Should be '2.5km **spacial** resolution'. - p.5, l.22: Remove comma between 'atmospheric model' and 'which is'. - p.6, l.18: The sentence 'This nudging is enabled only above the first local grid level over the surface' sounds a bit complicated. I suggest to write: 'This nudging is enabled starting from the second grid level above the local surface.' - p.8, l.14: Missing space between 'N_i' and the following word. - p.8, l.28: What is meant by '... at the level of the third-fourth floors'? Do you mean '...at heights between the third and fourth floor'? - p.9, l.19: '...junctions at DP and around (the sub-area 4)', better: '...junctions at DP (around sub-area 4)'. - p.11, l.10: Replace the semi-colons by comma and remove 'the' from the list. - p.11, l.11: 'Figure 8 shows that **the fireplaces**...' - p.14, l.20: This sentence is hard to understand. Please rephrase it. - Table 1: Entries are hard to read. Reduce the space between lines which belong to the same entry within the table to enhance readibility. - Table 2: The explanation below is wrong. Scenarios are not named as 'boh_...'. - Figure 3: Why is the figure caption written in italic? - Figure 5 and 7: The unit of the concentration does not use superscript for cubic-metres within the figure caption. - p.38, l.5: author names are written in capitals while all other entries are not.

---

## Referee Comment (RC2) · Anonymous Referee #2 · 28 Oct 2019

General comments: The manuscript "" written by Tobias Wolf describes the result from very high-resolution large eddy simulations of air pollutant dispersion in Bergen, Norway. The manuscript contains novel investigation to demonstrate the importance of resolving local circulations on detailed air quality assessment, especially in the inlet of the sea between steep slopes (i.e., fjord). The topic of the manuscript is certainly within the scope of ACP. Overall, the manuscript is well written and easy to follow. I would like to consider the publication of the manuscript from ACP, while I have several comments below which should be addressed before publication.

Specific comments:

Abstract Could you make novel findings more explicit?

[Figure]

3.1 The PALM Model The assumption that loss processes are omitted in the simulations seems to be plausible under the condition used in this study. But it is better to quantitatively show that typical time scales of these loss processes are sufficiently longer than typical transport time scale to justify the assumption for readers.

Is dry deposition process not considered?

4.1. Baseline scenario simulations p. 10, l. 10-11: "This was also confirmed with a detailed analysis of local pollutant measurements at the two reference" p. 10, l. 23-24: "This was visible in the PALM simulations as well as in the detailed analysis of the available air pollution observations at the tow reference stations." How did you confirm the dominant sources of NO2 and PM2.5 using measurement data and the consistency with the PALM simulation? Could you explain them briefly in the text or cite the references?

p. 10, l. 20-21: "Some urban areas might be affected by pollution transport over several kilometers and accumulation of emitted substances strongly different from the emission pattern." It is difficult to see the differences between concentration and emission patterns from Figures 1 and 5. Could you make emission flux maps used in the PALM simulations?

4.3. Air pollution pattern sensitivity to meteorological scenarios You show results from only three scenarios. How do you choose the three scenarios? You find no significant difference between the baseline simulation and other scenario simulations. Or These scenarios can be representative for all scenarios.

Technical correction:

p. 3, l. 28: please correct "air.pollution"

---

## Author Comment (AC1) · 5 Nov 2019

We would like to thank both referees for their help to improve this manuscript. Below is a list with point for point answers to both referees. The referee comments are repeated in italic font, eventual comments are given in normal font. Eventual changes to the text are explained in blue colour.

Referee #1

1) Page 5, line 12:

*Within the description of the data sets, it is mentioned that only supply vessels for the offshore oil industry are considered in the pollution emission of ships (p.5, l.9). This raises the question why not all ships are considered. There is no information given if these type of ships are the main pollutant emitter or if they contribute only by little to the total ship emissions. As the analysis of the contribution of ship emissions to the total air pollution is one of the major points of the manuscript, this question must clearly be answered within the text.*

We have explained qualitatively the relative impact of the supply-vessels compared to other relevant ship types.

2) Page 8, line 9 and page 11, line 29:

*During the presentation of the results in Sect. 4, the different used scenarios are mentioned at various points. Although, in Sect. 3.2.1, all considered meteorological conditions are mentioned and listed in Table 2, a list is missing which shows the actually conducted simulations with the used combinations of parameters. Such a table is also referenced on page 11, line 29 but is not part of the manuscript. Also, the naming of the scenarios is not explained in the text. An explanation is given below Table 2, but the explanation is incorrect and needs to updated.*

The mentioned reference to Table 1 was wrong and should have been a reference to Table 2. This is corrected in the text. We included a new paragraph into section 3.2.1 in order to explain the naming syntax and to explicitly state, what meteorological scenarios were conducted only as 12 h precursor runs and what scenarios were also conducted using the additional 6 h emission sensitivity simulations.

3) Page 12, line 26:

*In the discussion section, it is mentioned, that a resolution of at least 100m should be used to correctly resolve the diffusion processes of air pollutants within the Bergen region (p.12, l.26). I am missing a justification for this statement. Why can I also use 100m for such a study? Is there a gird-sensitivity study conducted which suggested a resolution of 100m or finer? A quotation is needed at this point. Otherwise, this has to be corrected to 10m resolution as this is the resolution used for this study.*

At 100 m or finer resolution, models begin to simulate turbulent eddies in the Kolmogorov inertial subrange. Coarser resolution eddies should thus be resolved explicitly, while finer resolution eddies can be parameterised.

We included a reference justifying the 100 m resolution threshold into the text.

4) *On page 10, line 26, it is mentioned that areas with higher density of fireplaces can easily be identified using Fig. 6 and Fig. 1. In my opinion, it is very hard to compare Fig. 1 and Fig. 6 because I cannot identify the coastlines in Fig. 6 (and also in Figs. 5-12). As I am not familiar with the area around Bergen, I need some help to navigate within the figures and the coastlines would drastically help. My suggestion is, either try to draw the coastlines into Figs. 5-12 or remove the white colour from the colour shading.*

We agree that the coastlines might not be visible at first glance. However, attempts to include coastlines in Figures 5-12 were problematic due to an overload of information in the figures. A

removal of the white colour shading would not help to mitigate this problem. An advantage of the chosen presentation with Google maps is the possibility to see the same map projection.

5) *- p.4, l.9: Should be '2.5km \*\*spacial\*\* resolution'. - p.5, l.22: Remove comma between 'atmospheric model' and 'which is'. - p.6, l.18: The sentence 'This nudging is enabled only above the first local grid level over the surface' sounds a bit complicated. I suggest to write: 'This nudging is enabled starting from the second grid level above the local surface.' - p.8, l.14: Missing space between 'N_i' and the following word. - p.8, l.28: What is meant by '... at the level of the third-fourth floors'? Do you mean '...at heights between the third and fourth floor'? - p.9, l.19: '...junctions at DP and around (the sub-area 4)', better: '...junctions at DP (around sub-area 4)'. - p.11, l.10: Replace the semi-colons by comma and remove 'the' from the list. - p.11, l.11: 'Figure 8 shows that \*\*the fireplaces\*\*...' - p.14, l.20: This sentence is hard to understand. Please rephrase it. - Table 1: Entries are hard to read. Reduce the space between lines which belong to the same entry within the table to enhance readibility. - Table 2: The explanation below is wrong. Scenarios are not named as 'boh_...'. - Figure 3: Why is the figure caption written in italic? - Figure 5 and 7: The unit of the concentration does not use superscript for cubic-metres within the figure caption. - p.38, l.5: author names are written in capitals while all other entries are not.*

Correction 1: The model is a spectral model. This should be indicated here. We corrected the description of the model resolution in the text.
We included all other corrections directly into the text.

Referee #2

1) Page 1, line 5:
*Abstract Could you make novel findings more explicit?*
We included two sentences describing the most relevant results for the pollution in Bergen.

Page 6, line 25:
2) *3.1 The PALM Model The assumption that loss processes are omitted in the simulations seems to be plausible under the condition used in this study. But it is better to quantitatively show that typical time scales of these loss processes are sufficiently longer than typical transport time scale to justify the assumption for readers.*
*Is dry deposition process not considered?*
The relevant time scales for the different mentioned pathways are up to today not entirely understood, especially for the cold winter-time temperature inversion conditions as they are found in Bergen (see for example the white paper of ALPACA, a planned large-scale campaign for the assessment of air pollution in cold climates: http://www.igacproject.org/publication/other-publications/alaskan-layered-pollution-and-chemical-analysis-alpaca-white-paper). A full description of these would therefore go well beyond the scope of this study.
Since the particles are treated as passive tracers, dry deposition is not considered, as mentioned in the paper.

3) *4.1. Baseline scenario simulations p. 10, l. 10-11: "This was also confirmed with a detailed analysis of local pollutant measurements at the two reference" p. 10, l. 23- 24: "This was visible in the PALM simulations as well as in the detailed analysis of the available air pollution observations at the tow reference stations." How did you confirm the dominant sources of NO2 and PM2.5 using*

*measurement data and the consistency with the PALM simulation? Could you explain them briefly in the text or cite the references?*

The analysis of the relevance of the different sources to the peak pollution levels in Bergen was based on the timing of the occurrence of peak pollution levels. The typical usage of wood-ovens is known to be during the evening. In addition, the timing of the rush-traffic with its distinct maxima in emissions in the early morning and late afternoon during workdays are known. This, together with the activity log in the harbour, and a relationship between the mean wind direction at different meteorological stations and the location of the emission sources and the reference stations can be used to assess the likely contribution of the three emission sources. The description of this is, however, well beyond the scope of this manuscript and therefore stated as "(not shown)". A reference is only available in a technical report and in Norwegian and therefore not of much help to international readers.

We included a short information into the text.

4) *p. 10, l. 20-21: "Some urban areas might be affected by pollution transport over several kilometers and accumulation of emitted substances strongly different from the emission pattern." It is difficult to see the differences between concentration and emission patterns from Figures 1 and 5. Could you make emission flux maps used in the PALM simulations?*

Emissions are indicated in Figure 1 and explained in section 3. The original data are stated in Table 1. A detailed emission map is not possible to present at such a high resolution in a format compatible with the journal requirements (e.g. emissions from roads could not be recognised on such a small map).

5) *4.3. Air pollution pattern sensitivity to meteorological scenarios You show results from only three scenarios. How do you choose the three scenarios? You find no significant difference between the baseline simulation and other scenario simulations. Or These scenarios can be representative for all scenarios.*

This is explained in section 4.3.

We included an improved description of the different meteorological scenarios and which were simulated including the 6 h emission simulations (see comments for referee #1, point 2).

6) *p. 3, l. 28: please correct "air.pollution"*

We corrected misprints in the manuscript.